# Study the muti-bolt fastening under different load positions in gear rack drilling rig

Jiangang Wang[1,2], Lei Shi[1,2], Ding Feng [1,2]*, Paul Tu[3]

**1** School of Mechanical Engineering, Yangtze University, Hubei, China, **2** Hubei Engineering Research Center for Oil & Gas Drilling and Completion Tools, Hubei, China, **3** Department of Mechanical and Manufacturing Engineering, University of Calgary, Calgary, Canada

☯ These authors contributed equally to this work.
* fengd0861@163.com

## Abstract

During the operation of drilling rigs, bolts are subjected to significant loads, rendering bolt groups vulnerable to failure due to uneven load distribution. This study investigates the multi-bolt load distribution characteristics of eight-gear and four-gear symmetric transmission mechanisms in drilling rigs. The spring stiffness method is utilized to analyze multi-bolt load distribution. A finite element model of the bolted connection is developed by integrating the structural shape and transmission mechanism. The model's accuracy is validated through a rack strain test under various loads. Based on the finite element analysis results, this study proposes an evaluation method for bolted connections using relative deformation difference. The impacts of bolt pitch, end distance, preload, and thickness of the derrick connection plate under different load positions on the connection are examined. This paper presents a methodology and conclusions that can inform the design of bolted connections for heavy-duty drive systems.

## 1 Introduction

In the current context of the highly competitive international oil and gas industry, conventional drilling rigs rely on the weight of the drill string to provide WOB (weight on bit) during drilling. They cannot actively apply WOB, resulting in low drilling efficiency in shallow wells due to low WOB. The gear-rack drilling rig is developed to achieve the lifting and running motion of the top drive unit through the gear and rack [1]. The gear-rack drilling rig has become a highly efficient drilling rig that many countries are competing to develop. The rig uses gear and rack drives as its primary transmission device. Due to the design of the transmission, the rack is subjected to heavy loads as a critical component [2, 3], which means that the reliability of the rack mounting affects the operation of the rig. Fig 1 shows how the rack is mounted in the rack drilling rig. In recent decades, bolted connections have remained widely used in modern rigs due to their ease of assembly and disassembly.

Eurocodes (Eurocode 3: Design of steel structures) and Standards of China (GB50017-2003) in developing the standard, both through the experimental way to study the parameters that affect the performance of bolted connection (bolt pitch, end distance and edge distance,

---

**Funding:** The paper is supported by China Major national science and technology projects

(2016ZX05038-002-LH001 from), National Natural Science Foundation of China (52204002). There was no additional external funding received for this study. The funders had no role in study design, data collection and analysis, decision to publish, or preparation of the manuscript.

**Competing interests:** The authors have declared that no competing interests exist.

**(a) Welding**　　**(b) Bolted Joints**

**Fig 1. Rack fastened method.**

etc.), to develop bolted connection design standards. As the study of bolted connections deepens, some scholars focus on the effect of bolt distribution on the connection. The reliability of bolted connections is ensured by determining the "critical bolts" through bolt load distribution [4]. Su et al. [5] investigated the effect of different bolt arrangements on the structural performance of concrete beams using experiments. Introducing the concept of " balanced failure point "can be used for rapid design. However, due to the limited number of tests, it is not possible to give a clear relationship between the " balanced failure point " and the bolts. Bolted connections are suitable for a wide range of applications, and it is not economically feasible to study them all experimentally, while the analytical method simplifies the actual engineering structure, resulting in a lack of extensiveness of the method [6]. Therefore, numerical simulation turns out to be a powerful and economical parametric learning tool. Like, Zhao et al. [7] and Zhang et al. [8] compared the 3D finite element model with the spring numerical analysis method to verify the reliability of their finite element calculations and investigated the effect of different fastening moments and bolt-hole clearances on multi-bolt load transfer using the finite element method. Li et al. [9] combined the spring method with finite element theory, based on which an improved spring method considering hole clearance and friction effects is proposed. Study the impact of different parameters of bolted connections on load distribution. Liu et al. [10] modified the stiffness model by considering the effect of bolt-hole deformation on the connection part. The improved stiffness model with the finite element method predicts the multi-bolt load distribution.

For the bolted connection problem, most scholars focus on the case of single-end load. Zhao et al. [11] investigated the effect of axial deviation load on bolt load distribution. However, it is impossible to make a reasonable design of the bolt arrangement by this method in the case of complex load-bearing conditions of the connected parts. In addition, some studies are assuming the bolts to be rigid, which makes the amount of energy dissipated by the bolt

deformation neglected. Belardi et al. [12–14] proposed a finite element method that incorporates beam and spring calculations to determine stiffness values for evaluating bolted connections. This approach considers factors such as friction, bolt preload, and radial clearance, and provides an analysis of the mechanical response in the bolted connection region. Sharos and McCarthy [15] proposed a method to study bolt load distribution and other conditions by finite elements to design multi-bolt connection structures. In his subsequent study, he proposed a mathematical analysis to solve the multi-bolt load distribution in the elastic range. The load position changes also have a significant effect on the load distribution [16–18], while cycling at different stress levels can lead to accelerated bolt failure [19]. Based on the importance and role of the rack in TEDR, the relevant design standard only has bolt fastening design requirements within a specific range, which makes the bolt fastening lack science in the actual workplace. Therefore, there is an urgent need to establish a high-precision and economical rack fastening model.

This paper examines the mechanical response of load distribution in multi-bolt connections under different load positions, with the impact of partial gear failure on transmission structures. A finite element model for the gear and rack multi-bolt connection is developed, integrating the structural forms of the transmission mechanism. Utilizing the finite element model calculation results, the relative deformation difference method is proposed for evaluating bolted connections. The research investigated the influence of bolt parameters on multi-bolt load distribution, in conjunction with the variation in rack load position. Methods and results presented in this study can be applied to analyze multi-bolt load distribution and design bolted connections for heavy-duty drive systems.

## 2 Spring stiffness method for the load distribution of bolt in rack fastening

### 2.1 Rack fastening model

The gear-rack drilling rig is characterized by active pressurization, modularity, and a high degree of automation. This study focuses on an asymmetric drive mechanism consisting of

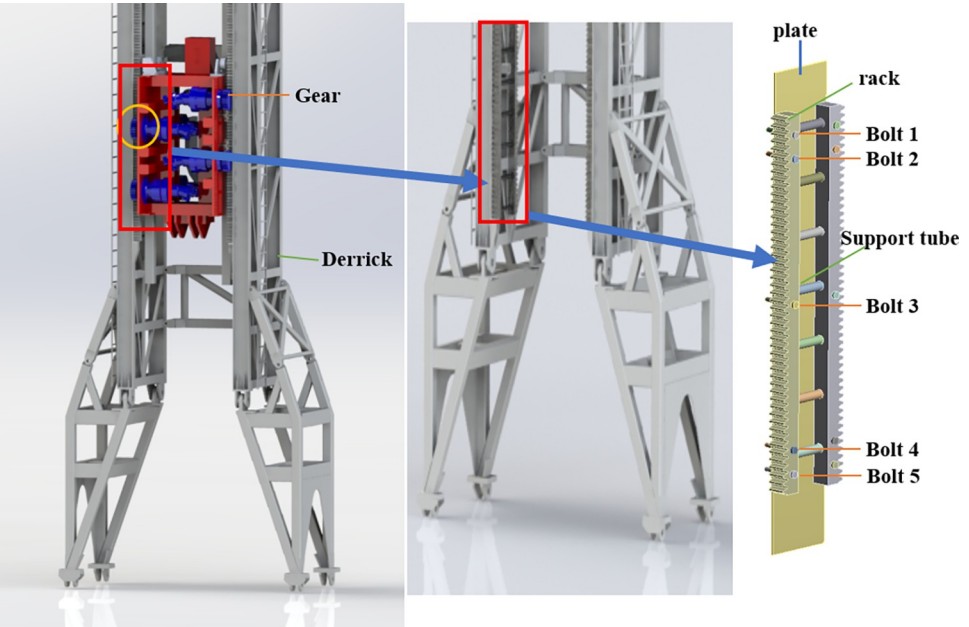

**Fig 2. Rack fastened geometry model.**

eight gears and four racks. The purpose of this mechanism is to lift the drilling string by engaging the gears and racks, thereby transmitting the drilling pressure. The rack-fixed model is the rack is fixed on the derrick connecting plate. The symmetrical structure of the mechanism and the operating conditions of the single rack gear are considered. To reduce the calculation time, the gear and bolt threads of the model are simplified. Fig 2 shows the rack fastening model.

## 2.2 Analytical model of bolt load distribution

The bolted connection model is idealized as a spring-mass system in the spring stiffness method as shown in Fig 3. The stiffness of the rack, the derrick connection plate and the bolts can be calculated according to [20, 21]. Static analytical equilibrium equations for each bolt and the connected parts are assembled based on the force analysis of each bolt node in the rack fastening model. The equilibrium equation matrix for the multi-bolt connection based on the spring method is obtained.

$$[K][X] = F \tag{1}$$

$$K = \begin{bmatrix} k_{spl5}+k_{sp4}+k_{b5} & -k_{b5} & -k_{spl4} & 0 & 0 & 0 & 0 & 0 & 0 & 0 & 0 \\ -k_{b5} & k_{b5}+k_{sk4} & 0 & -k_{sk4} & 0 & 0 & 0 & 0 & 0 & 0 & 0 \\ -k_{spl4} & 0 & k_{spl4}+k_{spl3}+k_{b4} & -k_{b4} & -k_{spl3} & 0 & 0 & 0 & 0 & 0 & 0 \\ 0 & -k_{sk4} & -k_{b4} & k_{b4}+k_{sk4}+k_{sk3} & 0 & -k_{sk3} & 0 & 0 & 0 & 0 & 0 \\ 0 & 0 & -k_{spl3} & 0 & k_{spl2}+k_{spl3}+k_{b3} & -k_{b3} & -k_{spl2} & 0 & 0 & 0 & 0 \\ 0 & 0 & 0 & -k_{sk3} & -k_{b3} & k_{sk2}+k_{sk3}+k_{b3} & 0 & -k_{sk2} & 0 & 0 & 0 \\ 0 & 0 & 0 & 0 & -k_{spl2} & 0 & k_{spl2}+k_{spl1}+k_{b2} & -k_{b2} & -k_{spl1} & 0 & 0 \\ 0 & 0 & 0 & 0 & 0 & -k_{sk2} & -k_{b2} & k_{sk2}+k_{sk1}+k_{b2} & 0 & -k_{sk1} & 0 \\ 0 & 0 & 0 & 0 & 0 & 0 & -k_{spl1} & 0 & k_{spl1}+k_{b1} & -k_{b1} & 0 \\ 0 & 0 & 0 & 0 & 0 & 0 & 0 & -k_{sk1} & -k_{sb1} & k_{sk1}+k_{sk5}+k_{b1} & -k_{sk5} \\ 0 & 0 & 0 & 0 & 0 & 0 & 0 & 0 & 0 & -k_{sk5} & k_{sk5} \end{bmatrix} \tag{2}$$

$$X = \begin{bmatrix} x_1 & x_2 & x_3 & x_4 & x_5 & x_6 & x_7 & x_8 & x_9 & x_{10} & x_{11} \end{bmatrix} \tag{3}$$

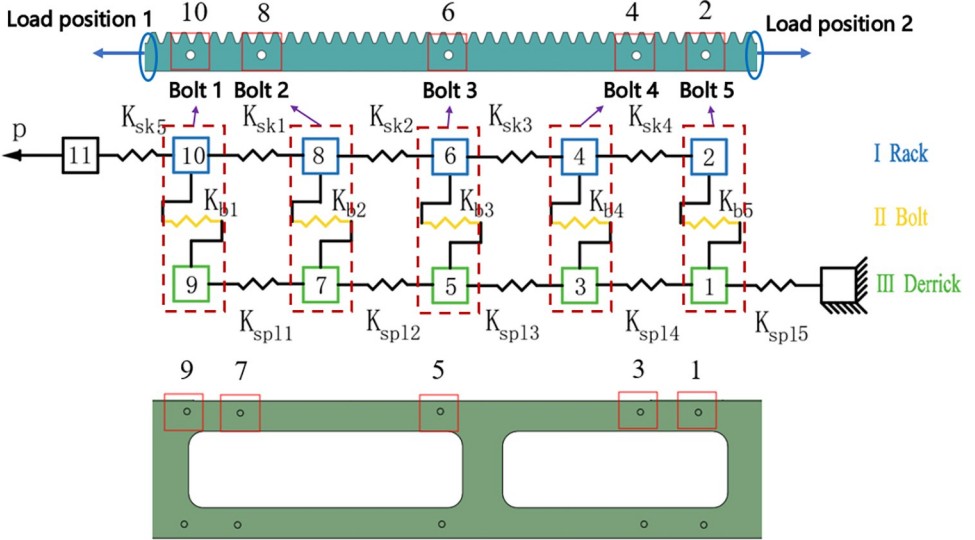

**Fig 3. Five-bolt spring-based model.**

$$F = \begin{bmatrix} 0 & 0 & 0 & 0 & 0 & 0 & 0 & 0 & 0 & 0 & P \end{bmatrix} \tag{4}$$

Where: $x_i$ is the deformation of mass i(i = 1–11), the stiffness of the rack is expressed by $K_{ski}$ (i = 1–5), the stiffness of the derrick connection plate is expressed by $Ks_{pli}$ (i = 1–5), the bolt stiffness is expressed by $K_{bi}$ (i = 1–5).

The bolt stiffness $K_b$, the connected part stiffness $K_{sk}$, and the $K_{spl}$ stiffness calculation formulas are represented by Eqs (5) and (6), respectively.

$$\frac{1}{K_b} = \frac{2(t_2 + t_1)}{3G_{BOLT}A_{BOLT}} + \left[ \frac{2(t_2 + t_1)}{t_2 t_1 E_{BOLT}} + \frac{1}{t_2 E_2} + \frac{1}{t_1 E_1} \right][1 + 3\beta] \tag{5}$$

$$K_{sk} = \left( \frac{Ewt}{p - d} \right) \tag{6}$$

Where $G_{BOLT}$, $A_{BOLT}$, and $E_{BOLT}$ are the shear modulus, cross-sectional area, and Young's modulus of the bolt, respectively, $t_1$, and $t_2$ represent the thickness of the rack and the thickness of the derrick connecting plate, respectively. $E$ is the modulus of elasticity, w is the width of the connected part, t is the thickness of the connected part, p is the bolt pitch, and d is the hole diameter.

## 2.3 Case study

The bolt arrangement is the symmetrical distribution of bolts on both sides (four in total) and one bolt in the center making a total of 5 bolts, taking a particular type of drill rig as an example for analysis. Structural parameters: bolt diameter is 36mm, bolt pitch at both ends is 200mm, bolt hole diameter is 37mm. The rack length is 3141.6mm, the thickness is 140mm, the derrick connection plate thickness is 16mm. The material parameters and properties are shown in Table 1.

## 2.4 Result analysis and discussion

This study aimed to investigate the effect of different bolt loads on the load distribution at different positions. The analysis specifically scrutinized the load distribution at the position 1 and position 2, as depicted in Fig 3. Despite the effect of friction between the rack and the derrick, which offsets the rack load, the load distribution between the bolts remains uneven. Fig 4 illustrates the decrease in the percentage bolt load as the distance from bolt 1 to the load position increases. The load distribution of bolts at different load positions changes obviously, and the obvious influence of load position on the load distribution of bolts is shown.

## 3 Rack bolt fastening model based on finite element method

The load distribution of the multi-bolt node connection is analyzed using the spring stiffness analysis method, which considers the influence in the two-dimensional direction. The effects

**Table 1. Material properties.**

| material | Elastic modulus (MPa) | Poisson's ratio | Tensile strength (MPa) | Yield strength (MPa) |
|---|---|---|---|---|
| Bolt | 205000 | 0.25 | 1080 | 900 |
| Rack | 212000 | 0.28 | 1080 | 930 |
| Plate | 209000 | 0.269 | 600 | 355 |
| Support tube | 206000 | 0.28 | 470 | 345 |

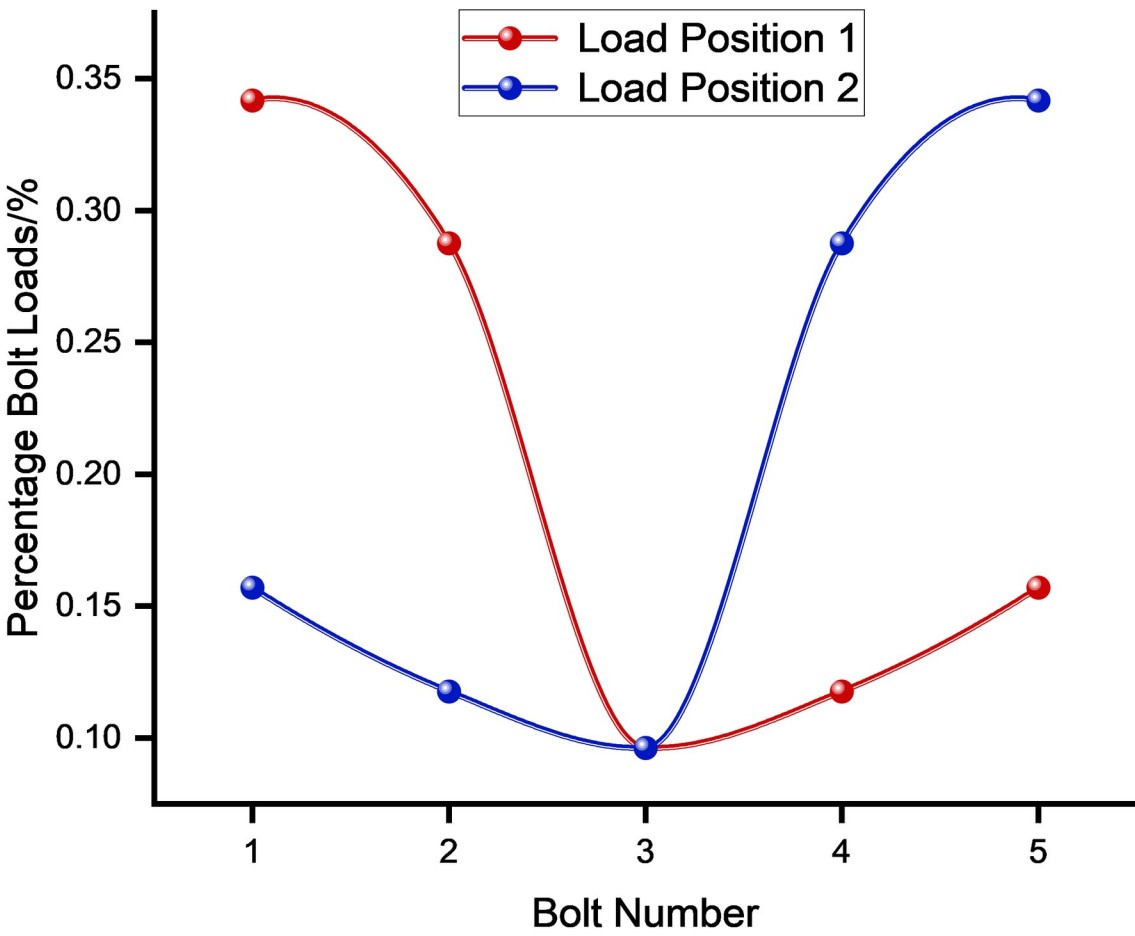

**Fig 4. Percentage bolt load in different positions.**

of load location, contact, and structural stiffness on anchor load distribution are analyzed. Three-dimensional finite element models of multi-bolt connections under different load positions are developed.

### 3.1 Finite element setting

**3.1.1 Element and Material properties.**   To improve computational accuracy and efficiency, a local refinement mesh and multi-scale mesh division method are implemented in the proximity of the rack bolt holes [22, 23]. The rack fastening model is constructed utilizing SOLID185 elements, which effectively capture the structural stiffness, contact and deformation characteristics. A detailed 3D finite element model of the specimen, as illustrated in Fig 5, is established, generating meshes with a Jacobian exceeding 0.7. The elastic-plastic and kinematic-hardening properties of the material are considered. The material properties of the bolt, rack, plate, and support tube are listed in Table 1.

**3.1.2 Interaction properties and boundary conditions.**   Contact between parts is modeled using CONTA174 or CONTA170 contact units [24], with "tangential contact" defined as frictional contact and coefficients of friction set to 0.2, 0.4, and 0.2, respectively. The "smoothing" option is employed for all contacts to improve the calculation accuracy [25, 26]. Fixed constraints are applied to the back of the connecting plate. The effect of preloading the bolt is

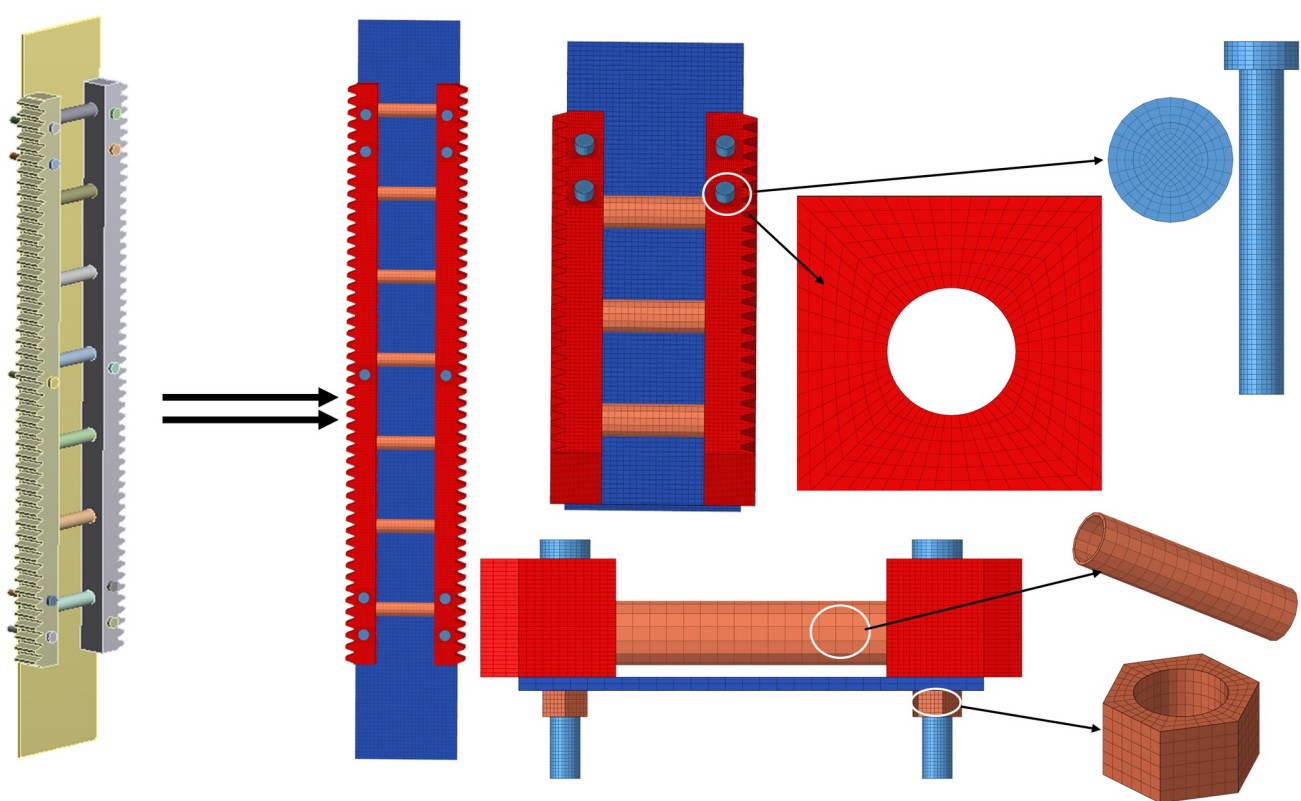

**Fig 5. Physical model and finite element model.**

achieved by applying a relative clamping force of 469 kN to the bolt and nut [22]. The simplified contact force between the gear and rack is 665kN loaded on the tooth face and 625kN in the Y direction. six load positions and directions of loading are marked on the rack as shown in Fig 6.

## 3.2 Model validation

Rack and pinion fasteners are tested at the lifting device test rig where static mechanical property tests are carried out to validate the numerical simulation results. Fig 7 shows the load device and test rig arrangement, where a cylinder is utilized instead of the drill string weight for the test. A strain measurement test is conducted at a single rack bolt hole, considering the symmetrical characteristics of the transmission system structure and strain sensor device's conditions.

Strain gauges for test loading are applied once the drive reaches the load position. This strategy prevents the drive from entangling with the data line during movement. Before the test, the collection program is loaded to 25T and subsequently unloaded to zero. This procedure verifies the normal functioning of the test equipment. Strain data is recorded using the SG802 instrument during the test, with a set collection time of ten seconds. This process is repeated thrice [27]. The test results obtained are then compared with the corresponding results from finite element calculations at the same load position, as depicted in Fig 8.

The results of finite element calculations and test data are compared. The strain values observed in the finite element of bolt holes are found to be marginally smaller than those recorded during the testing phase, a discrepancy that can be attributed to the relaxation of bolt

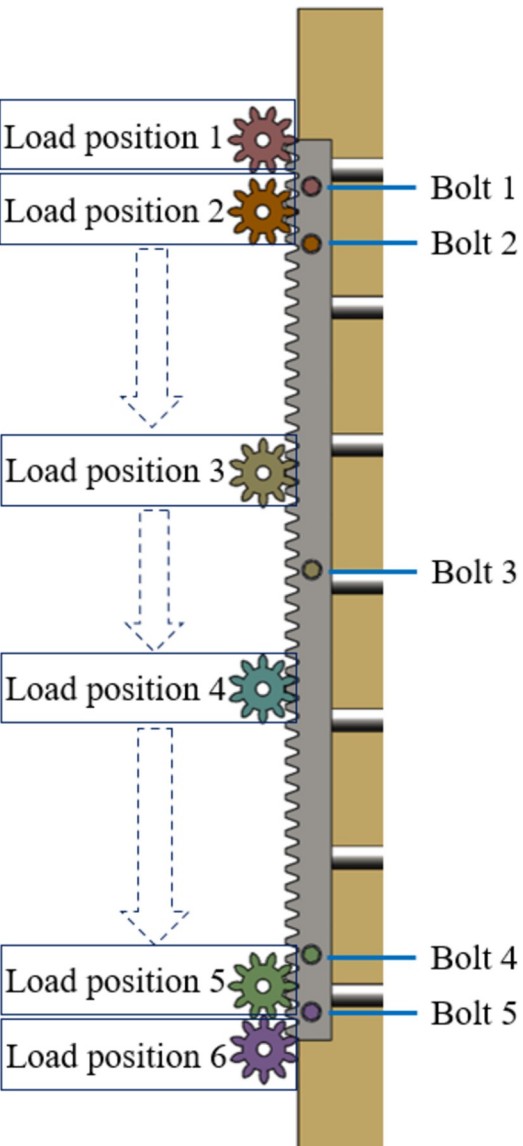

**Fig 6. The six positions of load on the rack.**

preload throughout the duration of the test. The reliability of the model is verified. Despite this small deviation, the strain values derived from the finite element results are in good agreement with the experimental results.

## 3.3 Finite element result

The effect of load position on bolt load distribution is investigated. Post-processing technology is used to calculate two measures: the mean deformation of the outer diameter at the junction between the bolt head and shank, and the mean deformation of the inner diameter of the rack bolt hole. Subsequently, the relative difference in deformation is calculated. The mechanical responses are analyzed under six different load positions, as illustrated in Fig 6.

The disparity in elastic deformation between the rack and the bolt under identical loading conditions can be attributed to the difference in stiffness inherent in their respective materials.

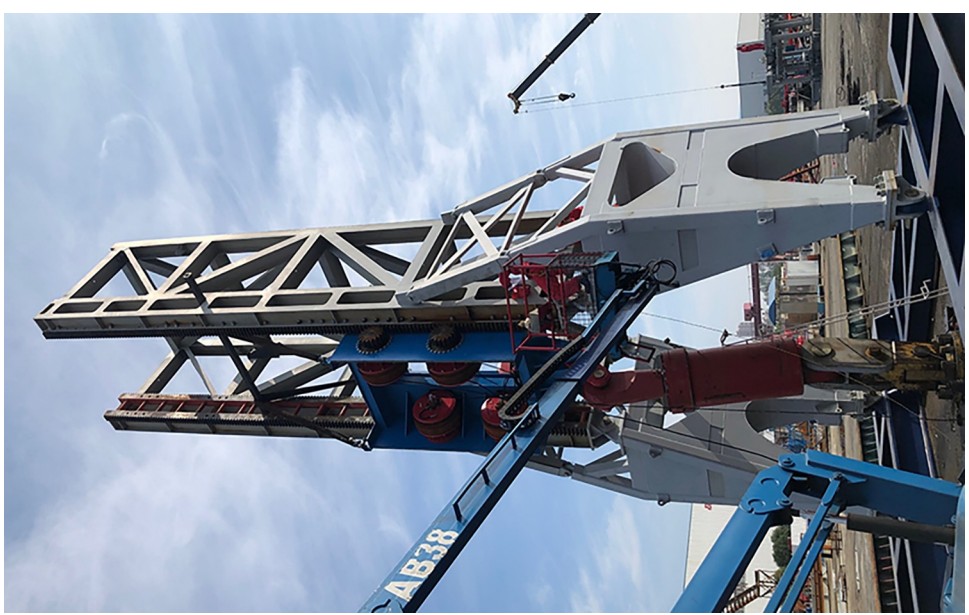

**Fig 7. Lifting device test stand.**

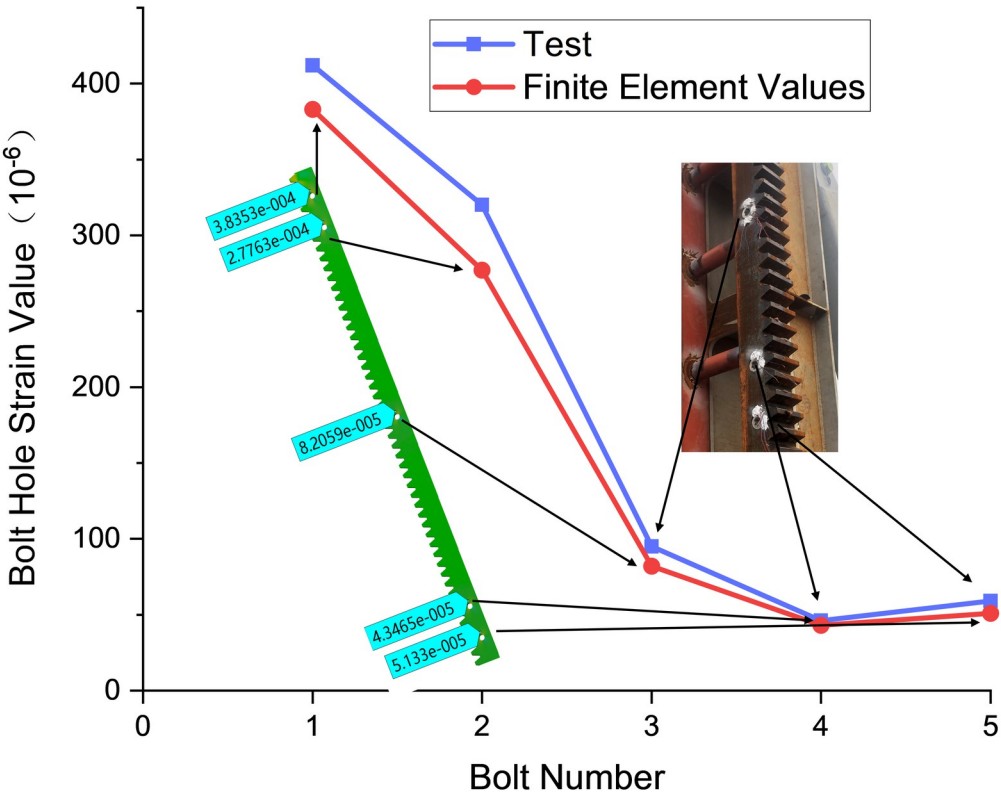

**Fig 8. Strain distribution along the cover rack bolt hole.**

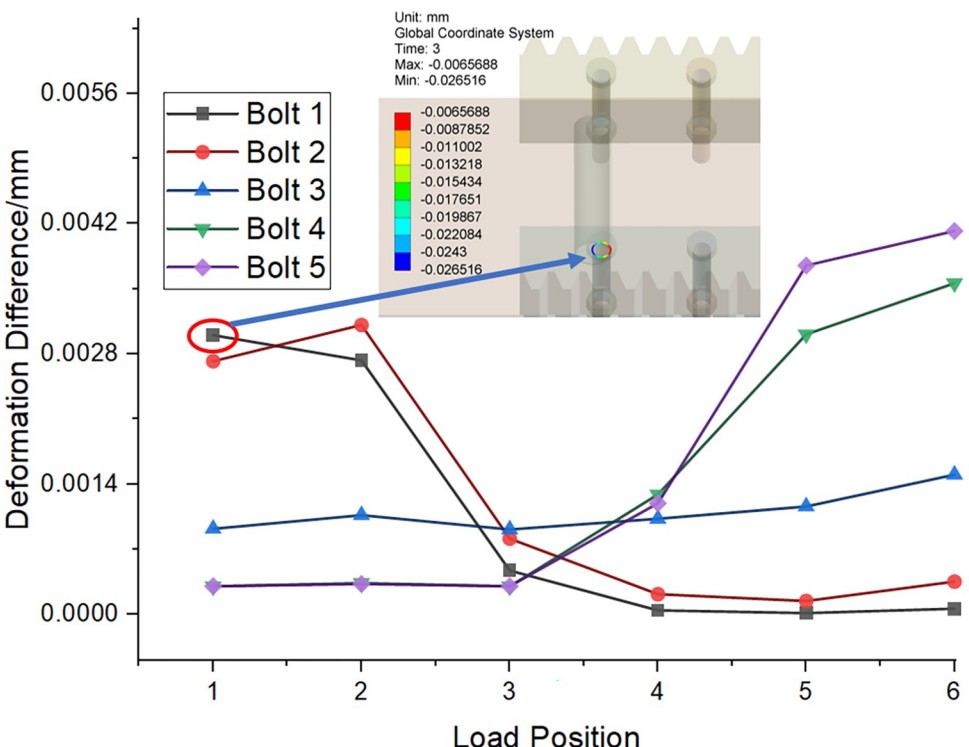

**Fig 9. Load position effect on bolt deformation difference.**

Fig 9 illustrates this phenomenon, especially for bolts 1 and 2 in loading positions 1 and 2. The effect of the difference in relative deformation is reflected in the closer the loading position is to the bolt. The results are proven to be consistent with previous studies that explored the effect of load position on bolt load distribution. Moreover, Fig 9 demonstrates that the end bolts, in comparison to the central bolt, are subject to more pronounced load variations. The geometrically centered symmetrical distribution of the rack is considered and the position of bolt 3 is minimal at different load positions. The difference in deformation at bolt 5 is higher than at bolt 1 which is caused by the downward load vector and the direction of gravity.

## 4 Parametric study

Based on the developed rack and pinion bolt fastening model, the effects of bolt spacing, bolt end distance and derrick plate thickness on bolt load distribution are investigated, as shown in Fig 10. In all the simulations, the default parameters for the finite element model are as follows, the bolt pitch = 200 mm, the bolt end distance = 160 mm, the connection plate thicknesses = 16 mm and load position as shown in Fig 6.

### 4.1 Comparative criteria

The effect of load position on bolt load distribution is described by theoretical analyzes and results of finite element calculations. Deformation extraction of bolts and rack and pinion bolt holes is used post-processing techniques which results in the effect of bolted connection parameters on fastening is investigated. The fluctuation of the deformation difference is used as an evaluation indicator of the effectiveness of bolt fastening at different load positions (a smaller standard deviation of relative deformation indicates better fastening). Consequently, the emphasis is placed on the trend of the data rather than value of the data.

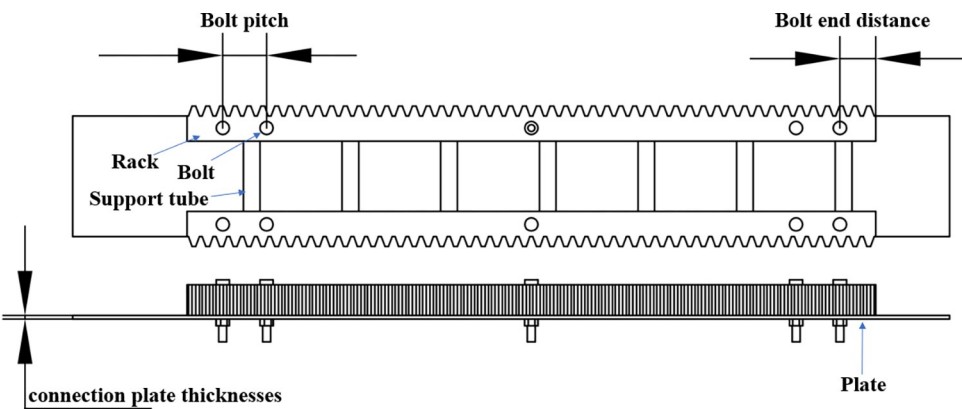

**Fig 10. Load position effect on bolt deformation difference.**

## 4.2 Effect of pitch

The effect of bolt pitch of 100, 150, 200, 250 and 300 mm on the deformation is studied, and different variations of other parameters are kept constant in the analysis. The curve of the effect of bolt pitch on the deformation difference at different load positions is shown in Fig 11.

**Fig 11. Bolt pitch effect on deformation difference under different load positions.**

Bolt pitch affects the value of bolt deformation fluctuation, but does not affect the varying law of bolt load distribution under different load positions due to Saint Venant's principle.

The mechanical response of bolts with different pitches at different load positions is compared. The results show that when the bolt pitch is increased from 100 mm to 300 mm, the fluctuation of the deformation difference decreases and then increases with a variation of 16.02%. The change in bolt pitch at different load positions resulted in a change in the load distribution of the bolt groups at both ends. According to the results, blindly increasing the bolt pitch leads to a decrease in the effectiveness of the rack fastening.

### 4.3 Effect of end distance

The effect of different bolt end distances (1.5d, 2d, 2.5d, 3d and 3.5d, where d is the bolt diameter of 36mm) on the fluctuation of bolt load distribution at different load positions is investigated. Fig 12 depicts the standard deviation for different bolt end distances under six load positions. The fluctuation of bolt load distribution decreases in amplitude at different load positions, with a maximum change of 12.96% when the bolt end distance increases from 1.5d to 3.5d. The results indicate that increasing the bolt end distance significantly improves the fastening effect of the bolt. When the bolt end distance is designed, the bending moment on the rack load is considered.

### 4.4 Effect of plate thickness

The effect of different plate thicknesses (16, 20, 25, 30 and 35 mm) on the fluctuation of bolt load distribution at different load positions is studied, as shown in Fig 13. The results show

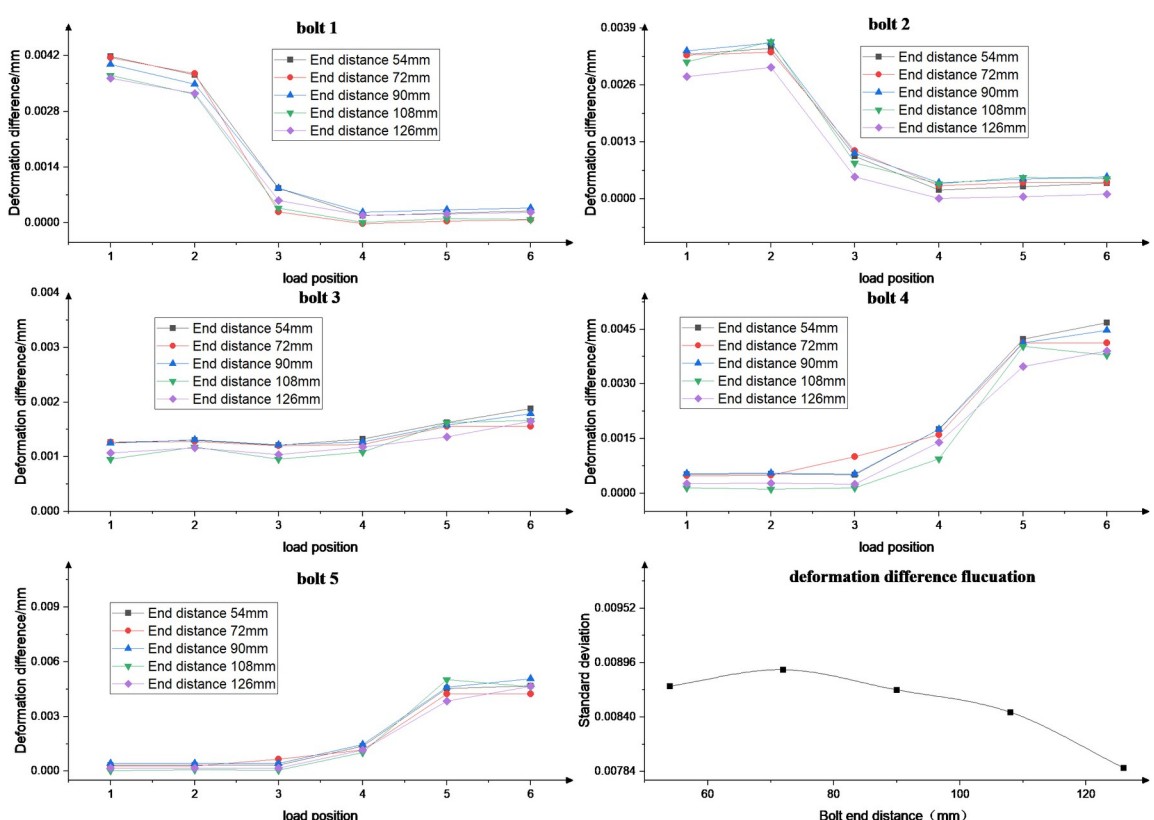

**Fig 12. Standard deviation for different bolt end distances.**

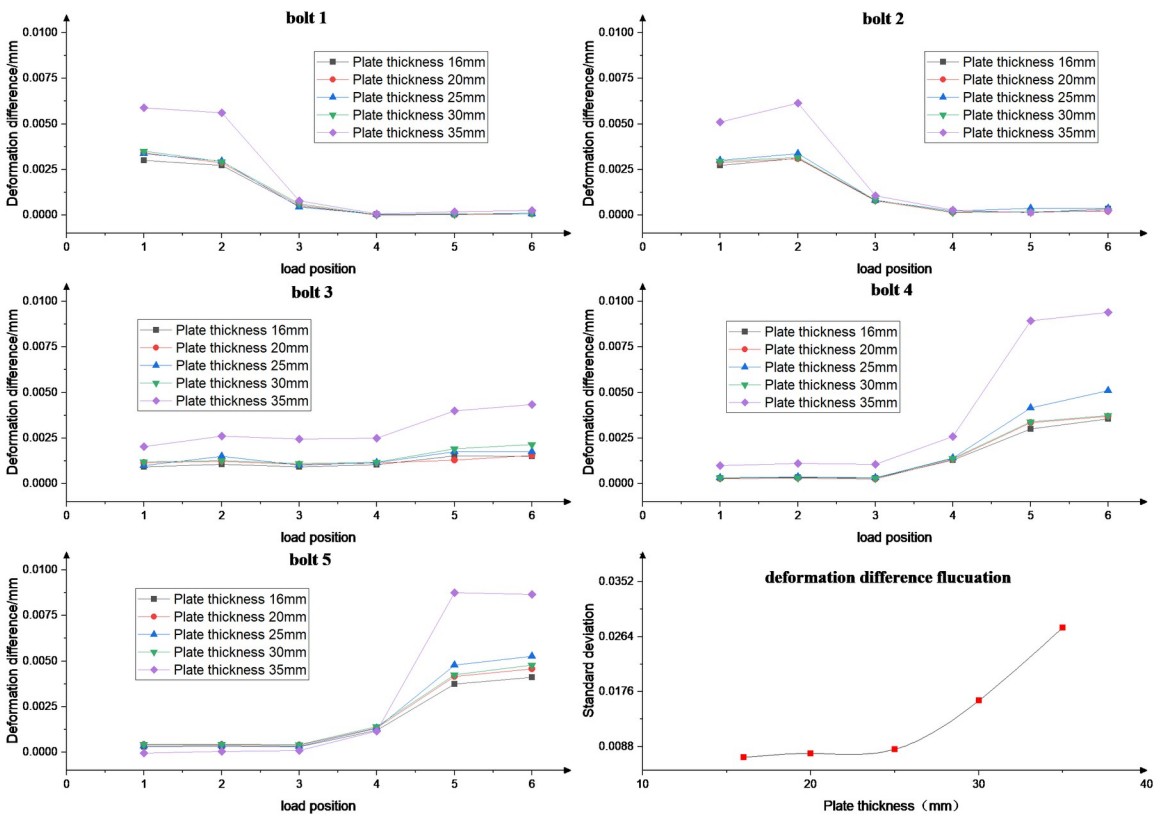

**Fig 13. Standard deviation for different plate thickness.**

that the fluctuation increases as the thickness of the derrick connection plate increases from 16mm to 35mm, for different load positions. Due to the increased stiffness of the structure resulting from the thicker plate, which leads to a reduction in the force transmission property and greater fluctuations under different load positions. The thickness of the derrick connecting plate is reduced in order to minimize the fluctuation of the deformation difference, while meeting the strength requirements.

## 4.5 Effect of preload

The effect of different bolt preload forces (409, 429, 449, 469 and 489 kN) on the fluctuation of bolt load distribution at different load positions is investigated, as shown in Fig 14. At different load positions, as the bolt preload increases from 409 kN to 489 kN, the fluctuation exhibits an approximately linear decrease.

The preload force is increased and the difference in deformation between the bolt and the rack is reduced. The load is resisted more significantly by the preload force, which improves the deformation difference caused by the material properties of the rack and the bolt. The preload force is increased, which improves the tightening of the bolt.

## 5 Conclusion

In this paper, the effect of load distribution on bolted connections of an eight-gear, four-rack symmetrical transmission mechanism is studied. A method for evaluating bolted joints based

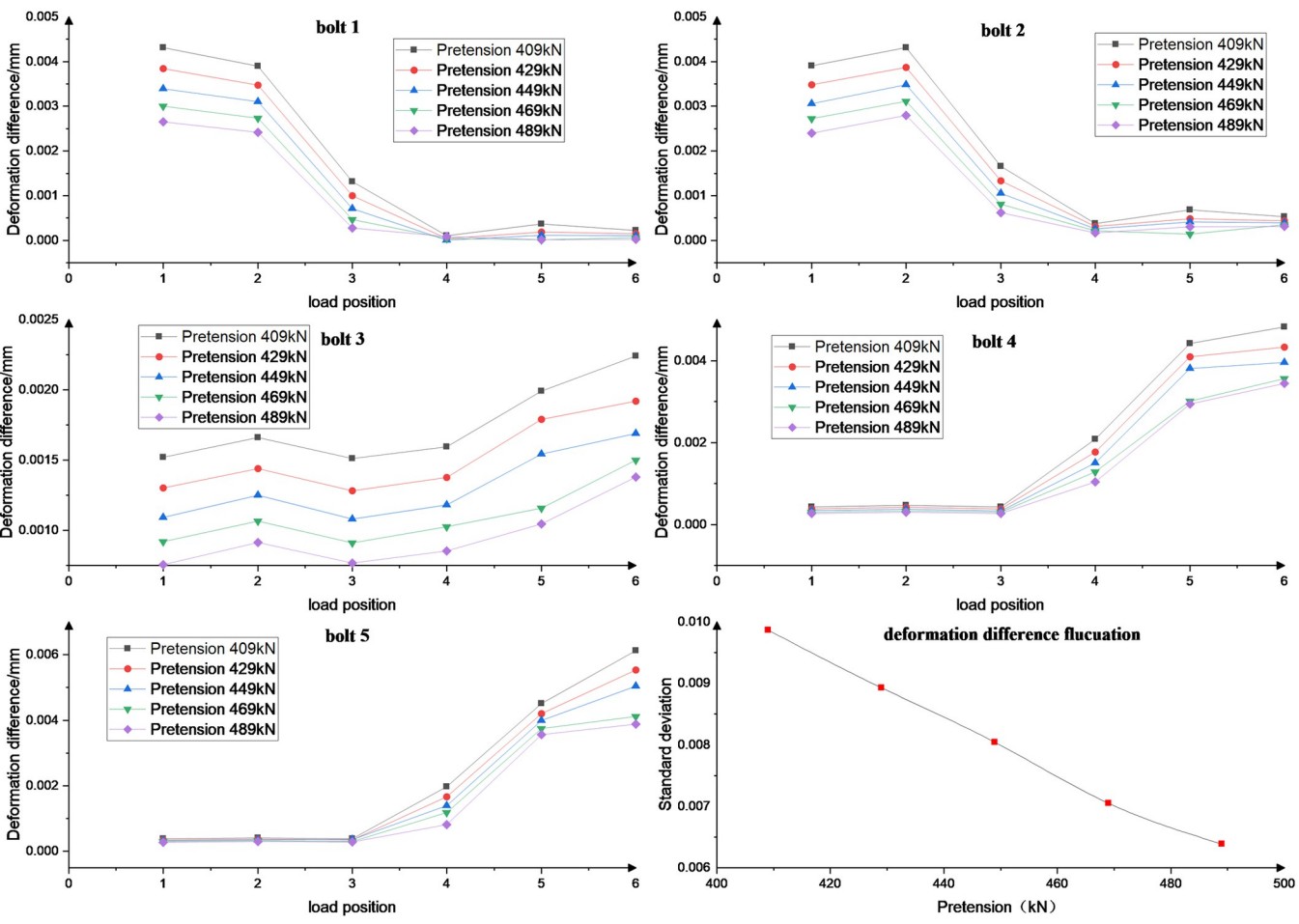

**Fig 14. Standard deviation for different preload.**

on relative deformation differences is proposed. The effect of bolt load distribution at different load positions and connection parameters is investigated. The main findings are as follows:

(1) Frictional multi-bolt connections exhibit a non-uniform distribution of bolt load, with the bolt nearest to the load position bearing the highest load. Upon conducting a spring stiffness analysis, the maximum difference in bolt load percentage under different load positions is quantified to be 59.1%.

(2) Finite element analysis elucidates that variations in relative deformation disparities act as a mechanical performance response to alterations in the position of the load. The reliability of this created finite element model is experimentally verified. There exists a robust correlation between the distribution of bolt load and the position of the load. In the design of multi-bolt connections for which the rack is fastened, the influence of the load position on the bolt arrangement method is considered.

(3) By utilizing the developed multi-bolt connection model and adjusting the bolt connection parameters, the bolt load distribution under different bolt connection parameters with different load positions can be accurately predicted. The proposed comparison criterion enables the optimal bolting parameters to be selected, and the design method is applicable to the design of heavy-duty rack multi-bolt connection parameters.

## Supporting information

**S1 File.**
(ZIP)

## Author Contributions

**Data curation:** Ding Feng.

**Funding acquisition:** Ding Feng.

**Methodology:** Jiangang Wang.

**Software:** Jiangang Wang.

**Supervision:** Paul Tu.

**Validation:** Lei Shi.

**Visualization:** Paul Tu.

**Writing – original draft:** Jiangang Wang.

**Writing – review & editing:** Lei Shi, Paul Tu.

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
