## [Decision Letter · Decision Letter 0]

16 May 2023

PONE-D-23-08274Study the muti-bolt fastening under various load positions in gear-rack drilling rigPLOS ONE

Dear Dr. feng,

Thank you for submitting your manuscript to PLOS ONE. After careful consideration, we feel that it has merit but does not fully meet PLOS ONE’s publication criteria as it currently stands. Therefore, we invite you to submit a revised version of the manuscript that addresses the points raised during the review process.

Please, address all the comments made by the reviewers. Please, also improve the English of the manuscript. 

We look forward to receiving your revised manuscript.

Kind regards,

Antonio Riveiro Rodríguez, PhD

Academic Editor

PLOS ONE

Journal Requirements:

"YES.Ding Feng is supported through grants 2016ZX05038-002-LH001 from Major National Science and Technology Projects."

"YES.Ding Feng is supported through grants 2016ZX05038-002-LH001 from Major National Science and Technology Projects."

6. We note that Figures 1 , 11 and 12 in your submission contain copyrighted images. All PLOS content is published under the Creative Commons Attribution License (CC BY 4.0), which means that the manuscript, images, and Supporting Information files will be freely available online, and any third party is permitted to access, download, copy, distribute, and use these materials in any way, even commercially, with proper attribution. For more information, see our copyright guidelines: http://journals.plos.org/plosone/s/licenses-and-copyright.

a. You may seek permission from the original copyright holder of Figures 1 , 11 and 12 to publish the content specifically under the CC BY 4.0 license. 

Reviewers' comments:

Reviewer's Responses to Questions

**Comments to the Author**

1. Is the manuscript technically sound, and do the data support the conclusions?

Reviewer #1: Yes

Reviewer #2: Partly

Reviewer #3: Yes

Reviewer #4: Yes

2. Has the statistical analysis been performed appropriately and rigorously? 

Reviewer #1: Yes

Reviewer #2: N/A

Reviewer #3: Yes

Reviewer #4: N/A

3. Have the authors made all data underlying the findings in their manuscript fully available?

Reviewer #1: Yes

Reviewer #2: No

Reviewer #3: Yes

Reviewer #4: Yes

4. Is the manuscript presented in an intelligible fashion and written in standard English?

Reviewer #1: Yes

Reviewer #2: No

Reviewer #3: Yes

Reviewer #4: No

5. Review Comments to the Author

Reviewer #1: 1. The paper is weird that the model validation is after the parametric analysis.

2. section 4.3 is too short to be a seperate section.

3. I do not believe the authors know how to write an academic paper, please refer to: https://doi.org/10.1016/j.jcsr.2018.12.020. You will find some useful information.

4. The matierial propertis, boundary conditions, interaction properties should be introduced in details.

5. The paper needs a major revision for sure, not only the content, but also the organization and language.

6. The quality of the figures also needs improvement.

Reviewer #2: The article in its current version is not suitable for publication. It is written in rather poor English. It is full of editorial errors. Bibliographic descriptions are missing. The authors use erroneous citations to items from the literature list.

No explanatory drawing of the adopted case study. Unintelligible Figures 4 and 7 (Which bolts are being referred to? What is their position?). Incomplete description of experimental studies.

Reviewer #3: This paper presents a study on multi-bolt transmission system of the drilling rigs. The topic is of great interests from both academic and industry. And the paper is well-written and therefore suggested to be considered to be published, with the following comments addressed:

1. Could the authors provide more details on the comparisons results of tests and numerical and theoretical predictions? It's critical to show the accuracy of the proposed predicting methods.

2. In the description to numerical modelling, could the authors please accordingly cite the references for the procedures that referring to some existing and well-known models in literature?

3. It's more logic to validate the developed numerical model first, then used it to conduct parametric studies. So, it's suggested to switch the order of sections in the current paper.

4. There are a few more relevant literature published recently. The authors may refer to them to get the updated ideas on the behaviour of high strength steel multi-bolt connections' failure in different failure modes, like staggered fracture, bearing and block tearing:

[1] Jiang, K., Zhao, O., Tan, KH. (2020). Experimental and numerical study of S700 high strength steel double shear bolted connections in tension. Engineering Structures, 225, 111175.

[2] Jiang, K., Tan, KH., Zhao, O. (2021). Net section fracture of S700 high strength steel staggered bolted connections. Thin-Walled Structures,164, 107904.

[3] Jiang, K., Tan, KH., Zhao, O., Gardner, L. (2021). Block tearing of S700 high strength steel bolted connections: Testing, numerical modelling and design. Engineering Structures, 246, 112979.

Reviewer #4: The paper proposes a finite element simulation method for gear-rack drilling rigs which present several bolted

joints. Some aspects regarding the modeling description and the validation of the results need to be

improved. Therefore, the revised version of the manuscript should consider the following amendments:

Section 1 – For greater clarity, report the expanded names for the acronyms ECCS and CSCS.

Figure 2 – Report in this figure all the names of the components mentioned in Section 2.1 and give a

closer view of the fasteners to help identify their position and details.

Figure 3 is not recalled in the text.

Eq. (5) – Explain the evaluation of Exx and Eyy. The method described in Refs. [17,24] is aimed at the

analysis of bolts connecting composite plates, hence the plates present different Young’s Modulus

along the x and y directions. Is in this case necessary to distinguish two different Young’s Moduli?

Section 2.4 – The distinction between load positions 1 and 2 is not clear. Describe more this aspect

and add a figure to support the description.

Section 3.2 – Give further details about the differential deformation between rack and bolt. How is it

measured and what are the technical implications?

Section 3.4 – It would be interesting to add a parametric study regarding the preload of the bolts.

Is it possible to compare some results of Section 3 with the analytical approach to give validation of

the numerical method?

The introduction should discuss other references about the FE analysis of composite bolted joints:

https://doi.org/10.1016/j.compositesb.2021.109378

https://doi.org/10.1016/j.compstruct.2020.112166

https://doi.org/10.1016/j.compositesb.2019.05.034

https://doi.org/10.1016/j.compstruct.2020.112770

https://doi.org/10.1016/j.compstruct.2020.113199

https://doi.org/10.1016/j.prostr.2020.02.078

The English of the manuscript needs to be carefully checked to correct misspells, typos. Some

sentences also need to be rephrased.

6. PLOS authors have the option to publish the peer review history of their article (what does this mean?). If published, this will include your full peer review and any attached files.

Reviewer #1: No

Reviewer #2: No

Reviewer #3: No

Reviewer #4: No

---

## [Author Response · Author response to Decision Letter 0]

26 Jun 2023

Reviewer #1: 

1. The paper is weird that the model validation is after the parametric analysis.

In order to make the paper structure more logical, the model validation section has been modified after the paper modeling calculation. The model validation part is placed in section 3.2 of the paper, after the Finite element setting and before the finite element result.

2. section 4.3 is too short to be a seperate section.

Due to the adjustment of the structure of the paper, the analysis results of the original section 4.3 have been placed in the model verification section.

3. I do not believe the authors know how to write an academic paper, please refer to: https://doi.org/10.1016/j.jcsr.2018.12.020. You will find some useful information.

The reference "https://doi.org/10.1016/j.jcsr.2018.12.020." has provided some writing ideas for my paper in terms of the structure of the paper and research philosophy, and has been rewritten and cited for the structure of the paper with the citation number "24".

4. The matierial propertis, boundary conditions, interaction properties should be introduced in details.

The finite element pre-processing settings such as element type settings, material properties, contact settings and boundary are described in sections 3.11 and 3.12.

5. The paper needs a major revision for sure, not only the content, but also the organization and language.

Thank you for your valuable and thoughtful comments. We have carefully checked and improved the English writing in the revised manuscript.

6. The quality of the figures also needs improvement.

The quality of the paper's pictures has been enhanced, particularly in terms of the structural form of the object under study, the composition of its parts, and the additional description of boundary conditions for calculations.

 

Reviewer #2: 

1. It is written in rather poor English. It is full of editorial errors. 

Thank you for your valuable and thoughtful comments. We have carefully checked and improved the English writing in the revised manuscript.

2. Bibliographic descriptions are missing. 

The bibliographic section has been added.

3. The authors use erroneous citations to items from the literature list.

The paper has carefully checked the problem of reference citations, and the problem of multiple citations for the same paper has been corrected.

4. No explanatory drawing of the adopted case study.

In the section on parameter learning, the thesis illustrates the parameters and their numerical values with this figure10 and word explanation.

5. Unintelligible Figures 4 and 7 (Which bolts are being referred to? What is their position?). 

The bolts and load positions are explained both in words and graphically, as shown in Figures 4 and 7.

6. Incomplete description of experimental studies.

The overall structural framework of the paper has been revised, and the original experimental analysis section has been modified to the validation of the finite element model, enhancing the description of the original experimental results.

 

Reviewer #3: This paper presents a study on multi-bolt transmission system of the drilling rigs. The topic is of great interests from both academic and industry. And the paper is well-written and therefore suggested to be considered to be published, with the following comments addressed:

1. Could the authors provide more details on the comparisons results of tests and numerical and theoretical predictions? It's critical to show the accuracy of the proposed predicting methods.

The Theoretical Calculations, Numerical Simulations and Strain Tests section provides further research details and explanations of boundary conditions, material properties and contact settings. The details are added in section 2.3 case study, section 3.11 Element and Material properties, section 3.12Interaction properties and Boundary conditions, 3.2 model validation.

2. In the description to numerical modelling, could the authors please accordingly cite the references for the procedures that referring to some existing and well-known models in literature?

For the creation of finite element model in numerical simulation, well-known model and literature are used in the simulation of relevant setting basis and evaluation method. The relevant references have been cited in the paper, numbered 25, 28 and 30.

3. It's more logic to validate the developed numerical model first, then used it to conduct para

---

## [Decision Letter · Decision Letter 1]

24 Jul 2023

PONE-D-23-08274R1Study the muti-bolt fastening under different load positions in gear-rack drilling rigPLOS ONE

Dear Dr. feng,

Thank you for submitting your manuscript to PLOS ONE. After careful consideration, we feel that it has merit but does not fully meet PLOS ONE’s publication criteria as it currently stands. Therefore, we invite you to submit a revised version of the manuscript that addresses the points raised during the review process.

Please, address all the comments provided by reviewer 2. 

We look forward to receiving your revised manuscript.

Kind regards,

Antonio Riveiro Rodríguez, PhD

Academic Editor

PLOS ONE

Journal Requirements:

Reviewers' comments:

Reviewer's Responses to Questions

**Comments to the Author**

1. If the authors have adequately addressed your comments raised in a previous round of review and you feel that this manuscript is now acceptable for publication, you may indicate that here to bypass the “Comments to the Author” section, enter your conflict of interest statement in the “Confidential to Editor” section, and submit your "Accept" recommendation.

Reviewer #1: (No Response)

Reviewer #2: (No Response)

Reviewer #3: All comments have been addressed

2. Is the manuscript technically sound, and do the data support the conclusions?

Reviewer #1: (No Response)

Reviewer #2: Yes

Reviewer #3: Yes

3. Has the statistical analysis been performed appropriately and rigorously? 

Reviewer #1: (No Response)

Reviewer #2: N/A

Reviewer #3: Yes

4. Have the authors made all data underlying the findings in their manuscript fully available?

Reviewer #1: (No Response)

Reviewer #2: Yes

Reviewer #3: Yes

5. Is the manuscript presented in an intelligible fashion and written in standard English?

Reviewer #1: (No Response)

Reviewer #2: No

Reviewer #3: Yes

6. Review Comments to the Author

Reviewer #1: (No Response)

Reviewer #2: The article looks much better, but still needs improvement. However, already in terms of minor changes, which include:

1. Remove the 'Content' section.

2. Use correct citations to items included in the literature list (for example, 'Su et al. [7]' instead of 'Su [7]' in line 56). This must be checked for all citations. Look at other articles on how this should be done, such as these: https://doi.org/10.3390/lubricants10050075 or https://doi.org/10.3390/ma15124354. References must exactly match the items given in the literature list. The citation in line 82, for example, is incorrect; instead of 'McCarthy[17]' it should be 'Sharos and McCarthy [17]'. Check the others.

3. The description for [13] in the literature list is incomplete. Check the others.

4. Correct the mathematical notations in lines 121 to 123. What exactly does i(i=e,1-4) mean?

5. Inform the reader what the bolt numbers in Figures 4 and 9 mean. Which bolts do these numbers refer to?

6. Extend the description of the FEM model. In what way were the bolts preloaded?

Reviewer #3: (No Response)

7. PLOS authors have the option to publish the peer review history of their article (what does this mean?). If published, this will include your full peer review and any attached files.

Reviewer #1: No

Reviewer #2: No

Reviewer #3: No

---

## [Author Response · Author response to Decision Letter 1]

1 Aug 2023

1. Remove the 'Content' section.

Reply: The 'content' section has been deleted.

2. Use correct citations to items included in the literature list (for example, 'Su et al. [7]' instead of 'Su [7]' in line 56). This must be checked for all citations. Look at other articles on how this should be done, such as these: https://doi.org/10.3390/lubricants10050075 or https://doi.org/10.3390/ma15124354. References must exactly match the items given in the literature list. The citation in line 82, for example, is incorrect; instead of 'McCarthy[17]' it should be 'Sharos and McCarthy [17]'. Check the others.

Reply: The reference citation format and the literature review author citation have been modified according to the reference (https://doi.org/10.3390/lubricants10050075) format.

3. The description for [13] in the literature list is incomplete. Check the others.

Reply: Incomplete references and problems with matches in the literature list have been corrected, as detailed in the paper's references.

4. Correct the mathematical notations in lines 121 to 123. What exactly does i(i=e,1-4) mean?

Reply: Thank you for improving the quality of the paper. The original symbolic representation is prone to ambiguity, and the formulae have been re-corrected and refined in Figure 3.

5. Inform the reader what the bolt numbers in Figures 4 and 9 mean. Which bolts do these numbers refer to?

Reply: In order to better illustrate the meaning of the bolt numbers, the position of the individual bolts has been supplemented in Figures 3 and 6.

6. Extend the description of the FEM model. In what way were the bolts preloaded?

Reply: Yes, the additions have been made to the original description in order to make the finite element model set-up description more detailed. Among them, regarding the loading method of the bolt preloading force, the expansion is completed and modified as follows: 'The effect of preloading the bolt is achieved by applying a relative clamping force of 469 kN to the bolt and nut.'”

---

## [Decision Letter · Decision Letter 2]

9 Aug 2023

Study the muti-bolt fastening under different load positions in gear-rack drilling rig

PONE-D-23-08274R2

Dear Dr. feng,

We’re pleased to inform you that your manuscript has been judged scientifically suitable for publication and will be formally accepted for publication once it meets all outstanding technical requirements.

Kind regards,

Antonio Riveiro Rodríguez, PhD

Academic Editor

PLOS ONE

Reviewers' comments:

Reviewer's Responses to Questions

**Comments to the Author**

1. If the authors have adequately addressed your comments raised in a previous round of review and you feel that this manuscript is now acceptable for publication, you may indicate that here to bypass the “Comments to the Author” section, enter your conflict of interest statement in the “Confidential to Editor” section, and submit your "Accept" recommendation.

Reviewer #1: All comments have been addressed

Reviewer #2: All comments have been addressed

Reviewer #3: All comments have been addressed

2. Is the manuscript technically sound, and do the data support the conclusions?

Reviewer #1: Yes

Reviewer #2: (No Response)

Reviewer #3: Yes

3. Has the statistical analysis been performed appropriately and rigorously? 

Reviewer #1: Yes

Reviewer #2: N/A

Reviewer #3: Yes

4. Have the authors made all data underlying the findings in their manuscript fully available?

Reviewer #1: Yes

Reviewer #2: (No Response)

Reviewer #3: Yes

5. Is the manuscript presented in an intelligible fashion and written in standard English?

Reviewer #1: Yes

Reviewer #2: (No Response)

Reviewer #3: (No Response)

6. Review Comments to the Author

Reviewer #1: (No Response)

Reviewer #2: The authors have corrected the article sufficiently. It remains to correct the title to: Study the multi-bolt...

Reviewer #3: (No Response)

7. PLOS authors have the option to publish the peer review history of their article (what does this mean?). If published, this will include your full peer review and any attached files.

Reviewer #1: No

Reviewer #2: No

Reviewer #3: No

---

## [Editor Report · Acceptance letter]

21 Aug 2023

PONE-D-23-08274R2 

Study the muti-bolt fastening under different load positions in gear rack drilling rig 

Dear Dr. Feng:

I'm pleased to inform you that your manuscript has been deemed suitable for publication in PLOS ONE. Congratulations! Your manuscript is now with our production department. 

Kind regards, 

on behalf of

Dr. Antonio Riveiro Rodríguez 

Academic Editor

PLOS ONE